# Comparisons of Chloroplast Genome Mutations among 13 Samples of Oil-Tea Camellia from South China

**DOI:** 10.3390/genes14051083

**Published:** 2023-05-14

**Authors:** Jing Chen, Kaibing Zhou, Xinwen Hu

**Affiliations:** 1Sanya Nanfan Research Institute of Hainan University, Sanya 572025, China; kimchen111@163.com; 2School of Life Science, Hainan University, Haikou 570228, China

**Keywords:** oil-tea camellia, cpDNA, SNPs, InDels, phylogenetic tree

## Abstract

The differences in cpDNA SNPs and InDels of 13 samples from single trees of different species or populations of oil-tea camellia in South China were examined in this study, and phylogenetic trees were reconstructed based on CDSs and non-CDSs of cpDNAs to research the evolutionary relationships among all samples. The SNPs of all samples included all kinds of substitutions, and the frequency of the transition from AT to GC was highest; meanwhile, the frequencies of all kinds of transversions differed among the samples, and the SNPs exhibited polymorphism. The SNPs were distributed in all the different functional regions of cpDNAs, and approximately half of all SNPs in exons led to missense mutations and the gain or loss of termination codons. There were no InDels in the exons of any cpDNA samples, except those retrieved from *Camellia gigantocarpa*, although this InDel did not lead to a frame shift. The InDels of all cpDNA samples were unevenly distributed in the intergenic region and upstream and downstream of genes. The genes, regions of the same gene, sites and mutation types in the same region related to the distributions of SNPs, and InDels were inconsistent among samples. The 13 samples were divided into 2 clades and 7 or 6 subclades, and the samples of species from the same sections of the *Camellia* genus did not belong to the same subclades. Meanwhile, the genetic relationship between the samples of *Camellia vietnamensis* and the undetermined species from Hainan Province or the population of *C. gauchowensis* in Xuwen was closer than that between *C. vietnamensis* and the population of *C. gauchowensis* in Luchuan, and the genetic relationship among *C. osmantha*, *C. vietnamensis* and *C. gauchowensis* was very close. In sum, SNPs and InDels in the different cpDNAs resulted in variable phenotypes among the different species or populations, and they could be developed into molecular markers for studies on species and population identification and phylogenetic relationships. The conclusion from the identification of undetermined species from Hainan Province and the phylogenetic relationships among 13 oil-tea camellia samples based on cpCDS and cpnon-CDS sequences were the same as those from the former report.

## 1. Introduction

Oil-tea camellia is a special oil tree belonging to the *Camellia* genus whose seeds contain rich oil and that was first planted in China long ago. This group of plants includes approximately 100 species, and there are 30 common species among them, such as *C. oleifera*, *C. meiocarpa* and *C. vietnamensis* [1]. Camellia seed oil contains abundant tea polyphenols and tea saponin, whereas olive oil does not have both components. Olive oil contains harmful components such as cholesterol and sinapic acid but camellia seed oil does not. Therefore, camellia seed oil has better nutritional and healthcare value, and the oil-tea camellia industry is expected to develop broadly [2]. Oil-tea camellia plants strongly resist many kinds of stress and widely adapt to many kinds of environments, showing extremely strong resistance to typhoons [3], so they can be planted in low-yielding or abandoned land and produce high yields, resulting in better ecological and economic benefits.

Oil-tea camellia was planted in Hainan Province approximately 2000 years ago. Approximately 6000 hm^2^ of oil-tea camellia forest was constructed in the 1960s–1970s, and 4600-year-old trees were found in Qionghai city. People in Hainan Province are accustomed to edible oil-tea camellia seed oil and view it as mysterious. As a result, the supply of oil-tea camellia seed oil cannot meet the market demand, so high-yield cultivars for afforestation need to be selected immediately to promote large-area afforestation at a high rate [4]. The indigenous species of oil-tea camellia trees in Hainan Province has been identified as *C. vietnamensis* based on full cpDNA sequences [5], but the absence of indigenous cultivars has become the first bottleneck in the development of the oil-tea camellia industry in Hainan Province [6].

The indigenous species of oil-tea camellia in Hainan Province is commonly mistaken for *C. oleifera* [7,8], and this is confirmed by the fact that the cultivars of *C. oleifera* outside the Hainan Island grow abnormally. Therefore, although the indigenous species of oil-tea camellia in Hainan Province has been identified as *C. vietnamensis* [5], it is still necessary to study the evolutionary relationships among *C. vietnamensis* from Hainan Province and the other oil-tea camellia species elsewhere in South China, which will aid in the selection of *C. vietnamensis* cultivars for afforestation and the collection of optimal germplasm of oil-tea camellia. Such research is also essential to establish a market for oil-tea camellia seed oil based on *C. vietnamensis* from Hainan Province and innovate and reasonably develop oil-tea camellia germplasm.

The identification of plants based on morphological characteristics is complicated by factors such as different environmental conditions and plant age, and identification based on DNA barcoding is limited in resolution among closely related species, which is not enough to differentiate them from each other [9,10]. The chloroplast genome (cpDNA) is called a “superbarcode” because it has some advantages compared with a single DNA barcode, including a smaller sequence, uniparental maternal inheritance without recombination, and much more abundant variation information, so it has begun to be used for the identification of different species and even different populations of the same species [11]. In cpDNA, the coding sequence (CDS) differs greatly in its evolution rate from the noncoding sequence (non-CDS), and there are low sequencing costs and small splicing errors. cpDNA is used to explore the systematic evolution, classification and identification of plant species more conveniently, accurately and cheaply than other materials [12]. Therefore, there have been many successful studies on the identification, classification and evolutionary relationship inference of plant species based on the method of cpDNA comparison in the past 30 years [13].

It is difficult to identify the species of *Camellia* plants because of hybridization and autopolyploidy [14]. Therefore, it is necessary to identify the different species and discuss their evolutionary relationships from multiple perspectives, including cpDNA comparative analysis. The NCBI genome database (https://www.ncbi.nlm.nih.gov/genome/organelle/, accessed on 8 Feburary 2022) contains 7556 full cpDNA sequences, including 45 from *Camellia* (with 10 cpDNA sequences published by our group), providing a set of resources for analysing the evolutionary relationships and identifying species of *Camellia* and other plants. In this paper, based on the CDSs and non-CDSs of cpDNA, the indigenous oil-tea camellia species from Hainan Province were re-identified, and the significance of SNPs and InDels in cpDNA in the evolution of species was uncovered. In addition, the feasibility of using cpDNA SNPs and InDels as molecular markers of genetic evolution is discussed.

## 2. Materials and Methods

### 2.1. Experimental Materials

Leaf samples for cpDNA sequencing were collected from various plants, including *C. gauchowensis* from Gaozhou city, Guangdong Province (HD01); *C. gauchowensis* from Luchuan County, Guangxi Zhuang National Autonomous Region (HD02); *C. gigantocarpa* (HD03), *C. polyodonta* (HD04), *C. meiocarpa* (HD05), *C. semiserrata* (HD06), *C. oleifera* (HD07), *C. osmantha* (HD08), *C. vietnamensis* (HD09) and an undetermined species from Qionghai city, Hainan Province (HD10); an undetermined species from Danzhou city, Hainan Province (HD11); an undetermined species from Wuzhishan city, Hainan Province (HD12); and *C. gauchowensis* from Xuwen County, Guangdong Province (HD13). According to assembly and comparison after cpDNA sequencing, the cpDNA sequences of the HD10–HD13 samples (i.e., 3 undetermined species from Hainan Province and *C. gauchowensis* from Xuwen County, Guangdong Province) were identical, and four emerged from the cpDNA of HD10 [5]. Therefore, 10 cpDNAs were used to analyse gene mutations in this study, whose data can be obtained from https://www.ncbi.nlm.nih.gov/genbank/, accessed on 8 Feburary 2022.

### 2.2. SNP and InDel Analysis

The chloroplast genome of *C. oleifera* mentioned above was used as the reference genome, and the reads were mapped back to the reference genome with the BWA software. Then, the reads from PCR duplication were deleted with the Picard Tools software. According to the mapping results and comprehensively considering factors such as data features, sequencing quality and experiment operation, the probability of every possible genotype was calculated based on the actual data with the software GATK UnifiedGenotyper. The genotype with the highest probability was selected and regarded as the genotype at a specific site in the sequencing unit. Then, a quality value was assigned to reflect the accuracy of the gene, and the sequence identity was determined. According to the sequence identity, the polymorphic sites in the reference sequence were screened and filtered. The main steps were as follows: the SAM file was converted into a BAM file, the BAM file was sequenced, PCR duplicates were identified, reads of PCR duplicates were deleted, reads whose mapping Q values were no more than 10 were filtered and the indexes were calculated. The realignments of the sequences near the InDels were mapped, and then the variations with high accuracy were gained by filtering the results of variant calling based on SNPs and InDels GATK. SNP and small InDel annotation was carried out according to the off-annotation information of the reference genome with the Hannover procedure. With regard to the SNP and small InDel sites in the CDS region, their influences on the translation of the protein were identified.

### 2.3. Construction of a cpDNA Phylogenetic Map

Sixty-five complete cpDNA sequences were obtained according to the method in a previous report [5], and 10 sample sequences were added for analysis. CDSs and non-CDSs of the 75 full cpDNA sequences were extracted for phylogenetic analysis. The phylogenetic tree was constructed according to the method in the same former report [5]. Every group was analysed using the software MAFFT (default parameters), and sequence pruning was performed using Gblocks (with the following parameters: -t, d, -b5, h). The phylogenetic tree was constructed using the software IQTREE and the outgroup was set as *Hartia laotica* (NC_041509.1).

## 3. Results and Analysis

### 3.1. Identification and Preference Analysis of SNP Mutation Types

The cpDNA genome of *C. oleifera* (HD07) was used as the reference sequence, and the frequency distribution of the different SNPs from the different samples is shown in Figure 1. All samples had all the kinds of base substitution mutations, and the sites of every kind of substitution mutation were different from each other among the samples. However, transition mutations from AT to CG were the most common and transversion mutations from CG to GC and from TA to AT were the rarest and second rarest, ranking 6th and 5th, respectively. The other kinds of substitution mutations, such as transitions between CG and AT or transversions between TA and GC, ranked in the middle, and their rankings changed among samples. In sum, the frequency of transition mutations was higher than that of transversion mutations, and the ratio of the former to the latter was less than 2:1, so the substitution mutations of oil-tea camellia are biased towards transition mutations. Meanwhile, the substitution mutations appeared to be unique and polymorphic among the oil-tea camellia species or populations.

### 3.2. SNP Genotyping

The identification and statistical results of SNP genotyping are shown in Table 1. The total homozygous and heterozygous SNP sites of the different samples differed from each other, which meant that either the polymorphisms existed in the numbers of different genotypes of cpSNPs among the different species or they existed in the different populations of the same species. In other words, every sample had unique cpSNP sites. All samples showed more homozygous SNP sites than heterozygous SNP sites, which indicated that the variation in cpSNPs was relatively stable. Generally, the more total SNPs there were, the greater the sequence difference from *C. oleifera* (HD07), so the similarity to *C. oleifera* (HD07) exhibited the following order: *C. polyodonta* (HD04), *C. meocarpa* (HD05), *C. gauchowensis* from Gaozhou (HD01), *C. gigantocarpa* (HD03), *C. vietnamensis* (HD09), *C. gauchowensis* from Luchuan (HD02), *C. gauchowensis* from Xuwen (HD13) and the undetermined species from Hainan Province (HD10–HD12), *C. osmantha* (HD08) and *C. semiserrata* (HD06). Generally, the more heterozygous SNP sites there were, the greater the difference in SNP complexity from *C. oleifera* (HD07), such that complexity exhibited the following order: *C. semiserrata* (HD06), *C. osmantha* (HD08), *C. gauchowensis* from Xuwen (HD13) and the undetermined species from Hainan Province (HD10–HD12), *C. gauchowensis* from Luchuan (HD02), *C. vietnamensis* (HD09), *C. gigantocarpa* (HD03), *C. gauchowensis* from Gaozhou (HD01), *C. meocarpa* (HD05) and *C. polyodonta* (HD04). In sum, all the above results indicated that cpSNPs could be used as molecular markers to identify the different species and populations of oil-tea camellia.

### 3.3. SNP Annotation

The cpSNP annotations of 13 samples are shown in Table 2. The specific sequence regions and the site numbers of SNP locations are shown. Although they are not listed individually in this paper, this cpSNP location information would be essential to guide further research on the regulation of gene expression, molecular breeding and cpDNA genetic engineering.

The SNPs in the different functional regions of cpDNAs were distributed unevenly. The sequences of the functional regions, according to the numbers of genes containing cpSNPs, were as follows: exons, upstream and downstream overlapping regions, upstream, downstream, introns and intergenic spaces. However, the kinds and numbers of specific related genes in the different regions led to different magnitudes of differences among the samples.

The samples could be divided into five groups on the basis of the number of cpSNPs in the sequences of functional regions. The first group included the samples of the undetermined species from Hainan Province (HD10–HD12), the population of *C. gauchowensis* from Xuwen (HD13) and *C. osmantha* (HD08), with the sequence types showing the following order: exons, intergenic spaces, downstream, upstream, downstream and upstream overlapping regions and introns. The second group included *C. vietnamensis* (HD09) and the populations of *C. gauchowensis* from Gaozhou and Luchuan (HD01 and HD02, respectively)*,* and the sequences showed the following order: intergenic spaces, downstream, exons, upstream, downstream and upstream overlapping regions and introns. The third group included *C. osmantha* (HD08), and the order was intergenic spaces, upstream, exons, downstream, downstream and upstream overlapping regions and introns. The fourth group included samples other than those of *C. semiserrata* (HD06), with the retrieved order being exons, intergenic spaces, downstream, downstream and upstream overlapping regions, upstream and introns. The last group included *C. semiserrata* (HD06), and the order retrieved for the sample was exons, intergenic spaces, downstream and upstream overlapping regions, downstream, upstream and introns. The samples of each group had similar sequence type orders based on the numbers of SNPs, but the differences in the numbers of SNPs among the specific functional regions varied in magnitude among the samples.

The genes and the sites of the cpSNPs in exons differed among samples, and these different genes and sites led to missense mutations and the gain and loss of termination codons. Additionally, all samples, except those of *C. semiserrata* (HD06), the undetermined species (HD10–HD12) and the Xuwen population of *C. gauchowensis*, which showed loss of the termination codon in the *petL* gene, had no mutations leading to loss of the termination codon but did show missense mutations and gain of the termination codon. All samples had 1–2 gains of termination codons, and the different genes and sites showing missense mutations and gain of termination codons led to variation in the magnitudes of differences, and the majority of these mutations were synonymous mutations. All of these results are summarized in Table 3.

In general, the SNPs of the different samples had some common characteristics, but they still showed differences among the samples.

### 3.4. SNP Comparisons among Samples

The samples were divided into different groups according to their evolutionary relationships, geographical relationships, and the characteristics of differences between important traits. Then, the samples were compared within each group, and the results are shown in Table 4.

#### 3.4.1. Comparisons among the Different Populations of *C. gauchowensis* and among *C. gauchouwensis*, *C. vietnamensis* and the Undetermined Species from Hainan Province

A total of 51 SNPs were consistent among the samples of the three populations of *C. gauchowensis* (HD01, HD02 and HD13), and nine in exons included a missense mutation of the *rbcL* gene and a gain of the termination codon of the *accD* gene. Seventy-six SNPs were consistent among the samples of the three populations of *C. gauchouwensis* (HD01, HD02 and HD13), *C. vietnamensis* (HD09) and the undetermined species from Hainan Province (HD10–HD12), and twenty-three of those distributed among exons included the missense mutation of the *rbcL* gene and a gain of the termination codon of the *accD* gene. The results of further comparison between the two groups indicated that all of the SNPs in the former were also present in the latter. The results of the analysis combined with Table 2 revealed that each sample of the different populations had unique SNPs beyond those displayed in Table 3. The SNPs in the intergenic space of the *rpl20* gene differed significantly among the populations, and the samples of the undetermined species from Hainan Province (HD10–HD12) and the population of *C. gauchowensis* from Xuwen (HD13) had more SNPs than those of the populations of *C. gauchowensis* from Gaozhou and Luchuan (HD01 and HD02). The first two had more regions of unique SNPs, especially the exons of the genes *psbC*, *psbJ*, *psbL*, *psbF*, *psbE*, and *petL*, which all exhibited unique missense mutations.

#### 3.4.2. Comparison among *C. osmantha*, *C. gauchowensis* and *C. vietnamensis*

*C. osmantha*, *C. gauchowensis* and *C. vietnamensis* were distributed in the same production area, and the differences in the SNPs among them are worth noting. There were 48 SNPs consistent with each other, and they included the same missense mutations of the *rbcL* gene and the same gain of the termination codon of the *accD* gene. The SNPs in the intergenic spaces were completely different, and there were few SNPs in the other functional regions of genes. Forty-five SNPs were identified in the comparisons of the samples of *C. gauchouwensis*, *C. vietnamensis* and the undetermined species from Hainan Province, so there were three different SNPs upstream of the *atpB* gene and the overlapping regions of the *petG* and *petL* genes and *psaJ* and *petG* genes, respectively. The results of the analysis combined with Table 2 revealed that the genetic relationship between *C. osmantha* (HD08) and the population of *C. gauchowensis* from Gaozhou (HD01) was farther than that between *C. vietnamensis* (HD09) and the population of *C. gauchowensis* from Gaozhou (HD01), and the genetic relationship between the population of *C. gauchowensis* from Gaozhou (HD01) and *C. osmantha* (HD08) was similar to that between *C. osmantha* (HD08) and *C. vietnamensis* (HD09).

#### 3.4.3. Comparisons among *C. gauchowensis*, *C. vietnamensis*, Undetermined Species from Hainan Province and *C. osmantha*

Twenty-one SNPs were consistent among the samples of the three populations of *C. gauchowensis* (HD01, HD02 and HD13), *C. vietnamensis* (HD09), the undetermined species from Hainan Province (HD10–HD12) and *C. osmantha* (HD08). In addition, these SNPs were also observed in the comparisons of samples from *C. osmantha*, *C. gauchowensis*, and *C. vietnamensis* and those from *C. gauchouwensis*, *C. vietnamensis* and the undetermined species from Hainan Province. These comparisons also revealed the same missense mutation of the *rbcL* gene and gain of the termination codon of the *accD* gene. There were no identical SNPs in the intergenic spaces, and there were few identical SNPs in the other functional regions of genes, so the cpSNPs of *C. osmantha* were different from those of the undetermined species from Hainan Province.

#### 3.4.4. Comparison between *C. giganticarpa* and *C. meiocarpa*

The fruits of *C. giganticarpa* (HD03) and *C. meiocarpa* (HD05) are the largest and smallest among the afforestation species, respectively, examined in this paper, so it was worth comparing their SNPs. There were 38 consistent SNPs, no identical SNPs in the introns and far fewer identical SNPs in the other functional regions of genes than in the comparisons of the samples from the three populations of *C. gauchowensis* (HD01, HD02 and HD13) and the three populations of *C. gauchowensis* (HD01, HD02 and HD13), *C. vietnamensis* (HD09), *C. osmantha* (HD08) and the undetermined species from Hainan Province, respectively. The same missense mutations were observed in the exons of the *matk*, *atpF* and *psbF* genes, and the same gain of the termination codon of the *psbF* gene were observed. Additionally, there were numerous missense mutations that differed from each other, revealing large differences in the cpSNPs. Moreover, the SNPs in the intergenic space of the *rpl20* gene were different.

#### 3.4.5. Comparison between *C. polyodonta* and *C. semiserrata*

The flowers of *C. polyodonta* (HD04) and *C. semiserrata* (HD06) are both red, so it was worth comparing the SNPs of these two species. There were 27 shared SNPs, fewer than in the comparisons mentioned above, but there were more identical SNPs in the exons (18), more than in the comparisons of the samples from the three populations of *C. gauchowensis* (HD01, HD02 and HD13). Moreover, the same missense mutations of the genes *rpoB*, *ndhJ* and *psbF* and gains of the termination codons of the genes *ndhJ* and *psbF* were observed. There were no identical SNPs in the intergenic spaces, and there were fewer identical SNPs in the other functional regions of genes.

#### 3.4.6. Integrative Comparisons

The results of the comparisons of SNPs among the samples of *C. giganticarpa* (HD03), *C. polyodonta* (HD04), *C. meiocarpa* (HD05) and *C. semiserrata* (HD06) revealed only 3 shared SNPs, namely, 2 in the intron of the *rps16* gene and 1 in the overlapping downstream region of the *accD* gene and upstream region of the *psaJ* gene. Yet, there were no identical SNPs in the other functional regions of the genes. The results of the comparisons among all 13 samples revealed no identical SNPs.

In general, the similarity among the samples of *C. vietnamensis* (HD09), the undetermined species from Hainan Province (HD10–HD12) and the populations of *C. gauchowensis* from Gaozhou and Xuwen (HD01 and HD13) was higher than that among the samples of the three populations of *C. gauchowensis* (HD01, HD02 and HD13), which indicated that *C. vietnamensis* (HD09), *C. gauchowensis* (HD01, HD02 and HD13) and the undetermined species from Hainan Province (HD10–HD12) might belong to different populations of the same species. The similarity of the SNPs among the different populations of the same species was higher than that among the different species, the difference among the SNPs in the intergenic spaces was higher than that in the regions within genes, the abundance and differences of the SNPs in the exons were higher than those in the introns and there were no identical SNPs among all samples.

### 3.5. Identification of the Kinds of InDels and Calculation of Statistics at the InDel Sites

The results for InDels of all samples are shown in Table 5. The numbers of total deletion and insertion sites of the different samples differed from each other, meaning that polymorphisms existed in the numbers of the different types of InDels among the species or populations of the same species. In other words, every sample had unique InDel sites. The ratios of the number of deletions to the number of insertions were all greater than one, suggesting that the frequency of deletions was greater than that of insertions. The ratios of the number of deletions to the number of insertions of the samples from the populations of *C. gauchowensis* from Gaozhou and Xuwen (HD01 and HD13), *C. vietnamensis* (HD09) and the undetermined species (HD10–HD12) were almost equivalent, and the ratios of the number of deletions to the number of insertions of the samples from *C. osmantha* (HD08) and *C. semiserrata* (HD06) were slightly higher than those from the populations of *C. gauchowensis* from Gaozhou and Xuwen (HD01 and HD13), *C. vietnamensis* (HD09) and the undetermined species from Hainan Province (HD10–HD12). However, those of the other three species (HD03, HD04 and HD05) and the population of *C. gauchowensis* from Luchuan (HD02) were much higher. Therefore, the cpDNA InDels tended to be deletions, and the relative abundance of deletions depended on the species or population of oil-tea camellia. In summary, all the above results indicated that cpDNA InDels could be used as molecular markers to identify different species and populations of oil-tea camellia.

### 3.6. InDel Annotation

The cpDNA InDel annotations of 13 samples are shown in Table 6. The specific sequence regions and the site numbers of InDels were mainly distributed in the downstream, upstream and intergenic regions, whereas fewer were distributed in the introns, and almost none were distributed in the exons. Although the InDels are not introduced one by one in this paper, their location information would be essential for further research on the regulation of gene expression, molecular breeding and cpDNA-based genetic engineering.

#### 3.6.1. Exon Regions

The *rpoC2* exons of all samples, except the sample of *C. gigantocarpa* (HD03), had a six-base TTAAAA deletion. The InDel in the exon of the *rpoC2* gene of *C. gigantocarpa* (HD03) did not lead to a frameshift, but it might lead to the loss of the codon of phenylalanine and the loss of the termination codon. This meant that the InDels in the exon would be eliminated by mortality induction; yet, a few InDels not causing a frameshift, and thus having no effect on the phenotype, could occur occasionally.

#### 3.6.2. Intron Regions

There were no InDels in the introns of the samples retrieved from *C. gigantocarpa* (HD03), *C. polyodonta* (HD04) and *C. meiocarpa* (HD05), but the introns of the *rps16* and *ycf3* genes contained InDels; the intron of the *ycf3* gene contained one insertion, i.e., T; and that of the *rps16* gene in the other samples, except those from the population of *C. gauchowensis* from Xuwen (HD13) and the undetermined species from Hainan Province (HD10–HD12), contained two deletions, i.e., CTTTTTC, and one InDel, i.e., TTTTCC. However, the samples from the population of *C. gauchowensis* from Xuwen (HD13) and the undetermined species from Hainan Province (HD10–HD12) contained only one deletion, i.e., CTTTTTC. As a result, there were fewer InDels in the introns of different oil-tea camellia species or different populations of the same oil-tea camellia species, but there were some differences in the numbers and kinds of InDels in the introns among the different species and among the populations of the same species.

#### 3.6.3. Upstream, Downstream and Intergenic Regions

The InDels were mainly distributed in the upstream, downstream and intergenic regions of some genes in all samples, and the sizes of the InDels of all samples ranged from 1 to 14 bps. Sizes from 1 to 7 were common; sizes of 11, 12 or 14 bps were uncommon; and sizes of 9, 10 and 13 bps were rare. Importantly, there were two insertions (AACTTTAAATTGAA and AATTGAAAACTTTC) made up of 14 bases in the downstream sequence of the gene *matk* and one insertion (ATTCCAGTAAAATG) made up of 14 bases in the overlapping regions upstream of the gene *petA* and downstream of the gene *cemA*. There was one deletion (TTTATTCAATCA) made up of 12 bps downstream of the gene *rps16*, and one insertion (ATATAAATAAA) of 11 bps and one deletion (TATGGTAATCCA) of 12 bps in the gene *rpl20*. Additionally, the intergenic spaces of the *rpl20* gene of all samples contained many insertions and deletions of different sizes. Although the genes, the functional regions of the genes in which the InDels were distributed and the number of InDels were not consistent among all samples, there were differences among species or populations. The similarities of the InDels in the samples from the three populations of *C. gauchowensis* (HD01, HD02 and HD13), *C. vietnamensis* (HD09) and the undetermined species from Hainan Province (HD10–HD12) were the highest, and the similarities of the samples mentioned above and *C. osmantha* (HD08) were also high.

Table 6 shows only the genes, the regions of the genes containing InDels and the numbers of InDels; there were also differences in the sites, type of mutation (insertion or deletion), size and base composition, among other features. For example, there was one deletion (AAATAGAAATGAAATTTTTTT TATTTTATTAATAA) of 35 bps upstream of gene *rps16* in the sample of *C. meiocarpa* (HD05) and one insertion (TCTAAATAGAATTAGTATT) of 19 bps in the overlapping region upstream of the gene *ycf3* and downstream of the gene *rps4* in the sample of *C. polyodonta* (HD04). Only one of all samples contained both InDels, so both might be specific to the two species.

#### 3.6.4. Example Analysis of InDels

The results placing InDels in the intergenic space of the *rpl20* gene are shown as an example in Table 7. All samples contained the deletion TATAATA composed of 7 bp. The samples from the populations of *C. gauchowensis* from Gaozhou (HD01) and Xuwen (HD13), the undetermined species from Hainan Province (HD10–HD12) and *C. vietnamensis* (HD09) had the same InDels, and the AATAT and T deletions were the most common InDels among them; yet, the samples from *C. gauchowensis* from Luchuan (HD02) and *C. osmantha* (HD08) had more InDels than the four mentioned above, such as the insertion of AT and the deletion of TCATGAT in HD02 and the two insertions of T and the deletion of TATGGTAATCCA in HD08. Moreover, the additional InDels in HD02 and HD08 distinguished the two; the additional InDels of HD02 could separate HD02 from the populations of *C. gauchowensis*, and the additional InDels of HD08 could separate HD08 from *C. gauchowensis* and *C. vietnamensis*. However, each of the samples from HD02 to HD08 had a unique combination of the different InDels, such as the insertion of AT and the deletion of TCATGAT in HD02; the insertion of A and the deletion of A in HD03; the insertion of AATAG and A and the deletions of CTTAC and TCTTTT in HD05; the insertions of AT and T and the deletion of TATGGTAATCCA in HD06; and the two insertions of T and the deletion of TATGGTAATCCA in HD08. In summary, the insertion of A in HD03 and the insertion of AATAG and the deletions of CTTAC and TCTTTT in HD05, were the most important because they were the unique InDels of each sample, and the deletion of TATGGTAATCCA in HD06 and HD08 was more important because it did not exist in the other samples and could separate HD06 and HD08 from each other and the other samples when used in combination with the other unique InDels. Additionally, the deletion of AATAT and the two deletions of T were more important InDels because they could be used to distinguish *C. gaochowensis*, *C. vietnamensis*, *C. semiserrata* and *C. osmantha* from the other species.

The other InDels in the cpDNAs of all samples are not introduced individually here, but the characteristics of the differences in the InDels could be summarised based on results such as those presented above. The cpInDels were mainly distributed in the non-CDSs of cpDNAs, and the genes, functional regions, number, sites, type (insertion or deletion), size and base construction of the cpDNA InDels changed depending on the oil-tea camellia species or population. In general, the cpInDels of different samples shared some common characteristics but they still exhibited polymorphism.

### 3.7. Phylogenetic Inference

*H. laotica* was taken as the outgroup, and cpDNAs of all samples and seven other *Camellia* species were applied for phylogenetic reconstruction by maximum likelihood (ML) based on the CDSs. The results are shown in Figure 2. It divided all samples and the sequences from the NCBI into 2 clades; one included 2 subclades, and the other included 5 subclades. The subclade of *C. granthamiana* included samples of *C. vietnamensis* (HD09), *C. gauchowensis* (HD01, HD02 and HD13), the undetermined species from Hainan Province (HD10–HD12), *C. osmantha* (HD08) and *C. granthamiana*. The subclade of *C. azalea* included the samples of *C. semiserrata* (HD06) and *C. azalea*. The sample of *C. gigantocarpa* (HD03) formed an independent subclade. The subclade of *C. japonica* or *C. chekiangoleosa* included the samples of *C. polyodonta* (HD04), *C. japonica* and *C. chekiangoleosa*. The subclade of *C. oleifera* included the samples of *C. oleifera* (HD07) and *C. japonica* (the serial number from the NCBI differed from the one in the above subclade). The subclade of *C. sasanque* included the samples of *C. sasanque* and *C. meiocarpa* (HD05). *C. crapaelliana* formed an independent subclade. The samples of *C. gaochauensis* and *C. vietnamensis* are notable in that they showed the cluster nodes of the samples from the population of *C. gauchowensis* from Gaozhou (HD01) and *C. vietnamensis* (HD09) in the outermost position. Then, the cluster nodes of these two and the samples of the undetermined species from Hainan Province (HD10–HD12) or the population of *C. gauchowensis* from Xuwen (HD13) were located in the second outermost position, with the next node associated with the samples from the population of *C. gauchowensis* from Luchuan (HD02) and *C. osmantha* (HD08). Therefore, the phylogenetic relationships among *C. vietnamensis, C. gaochauensis* and the undetermined species from Hainan Province were closer than that between the populations of *C. gauchowensis* from Gaozhou and Luchuan, meaning that *C. vietnamensis*, *C. gauchowensis* and the undetermined species from Hainan Province could be merged into the same species, with *C. osmantha* very closely associated with them. Through further analysis involving combining the above results, the different SNPs from the different samples were used as molecular markers for the identification of species, populations and phylogenetic relationships based on CDSs because there were more SNPs and almost no InDels in exons of cpDNAs.

*H. laotica* was taken as the outgroup, and the cpDNAs of all samples and the seven other *Camellia* species were applied for phylogenetic reconstruction via ML methods based on the non-CDSs. The results are shown in Figure 3. The results were similar to those based on CDSs, and generally showed the same phylogenetic relationships. The difference from the phylogenetic trees based on CDSs was that the sample of *C. gigantocarpa* (HD03) was merged with the subclade of *C. crapaelliana* rather than an independent subclade, so the second clade included 4 subclades instead of 5 clades, making the phylogenetic relationship between the sample of *C. gigantocarpa* and the sequence of *C. crapaelliana* closer. In regard to the subclade of *C. granthamiana*, the phylogenetic trees were extremely similar to those based on CDSs, proving that *C. vietnamensis*, *C. gaochauensis* and the undetermined species from Hainan Province belonged to the same species and that *C. osmantha* was closer than other species to them in terms of genetics. Through further analysis involving combining the above results, the SNPs and InDels from the different samples could be used as molecular markers for the identification of species, populations and phylogenetic relationships based on non-CDSs because there were some SNPs and InDels in the intergenic, upstream, downstream and intron regions of cpDNAs, and these SNPs and InDels were polymorphic among species and populations.

## 4. Discussion

### 4.1. Primary Analysis of the Genetic Mechanism Underlying the Phylogenetic Relationships within the Camellia Genus

As shown in this paper, some exons in the cpDNA of all samples contained SNPs, and the SNPs appeared to be polymorphic among the different samples. There were inconsistent synonymous mutations, missense mutations and stop-gain mutations among the different samples, and a few samples, such as that of *C. semiserrata* (HD06), the undetermined species from Hainan Province (HD10–HD12) and the population of *C. gauchowensis* (HD13), even contained one stop-loss mutation. Because the SNPs in the exons or the regions for gene expression regulation of all samples could lead to variation in the expression of some traits, the phenotypes varied among the different populations of *C. gauchowensis* and the different oil-tea camellia species. This phenomenon has been reported for nuclear genomes and chloroplast genomes. Synonymous codon usage bias results from synonymous SNPs [15], and the phenotype varies because of differences in gene expression depending on usage frequency and expression efficiency of the different synonymous codons [16]. The *Brassica pekinensis* mutation cer1, associated with the absence of wax, is a transition SNP of CT in the fourth exon of the *BraA09g066480.3C* gene and it leads to a missense mutation. Cabbage without wax resulted, so the phenotype changed greatly [17]. Two SNPs in the granule-bound starch synthase I gene of *Oryza sativa* resulted in great changes in the structure and the function of the protein, causing the rice quality to change significantly [18]. SNPs in the tetrahydrocannabinolic acid (THCA) synthase gene could influence the strength of toxicity of *Cannabis sativa*, which allowed hemp products to be divided into those with strong toxicity and weak toxicity based on the difference in SNPs and to be distinguished from other plants’ products [19]. In sum, cpSNPs had certain effects on the evolution of the different populations of *C. gaochauensis* and the different oil-tea camellia species.

According to the results in this paper, all samples, except for the sample of *C. gigantocarpa* (HD03), had no InDels in the exons of the cpDNAs, and *C. gigantocarpa* (HD03) had one deletion of 6 bp without a frameshift mutation. This InDel led to the loss of the amino acid and the gain of a termination codon, so it might have significant effects on the phenotype, making *C. gigantocarpa* different from the other plants of the *Camellia* genus. This result indicated that the InDels in the exons of cpDNA were responsible for variation among the populations and species if they led to missense mutations or nonsense mutations. However, cases of InDels in exons contributing to the phylogenetic relationships of organisms have not been reported, possibly because InDels cause frameshift mutations and result in serious harm and even mortality. The results in this paper showed that the InDels of the different samples were distributed in the regions of gene expression regulation, including the upstream, downstream and intergenic regions, even in the introns. Given that the mutation in the upstream promoter of the granule-bound starch synthase I gene of *Solanum tuberosum* could influence the expression of the gene [20], the InDels in the non-CDSs of cpDNA uncovered in this paper may also influence the phenotype by affecting gene expression, causing them to have certain effects on the evolution of *Camellia* plants.

### 4.2. Application Value Analysis of SNPs and InDels in the cpDNAs of Oil-Tea Camellia

The results in this paper reveal the following main points. The SNPs and InDels of cpDNAs were mainly distributed in the intergenic regions and the sequences upstream and downstream of genes and minorly in the introns, and the SNPs were still mainly distributed in the exons. Meanwhile, the CDSs of the samples contained different SNPs, and the non-CDSs of the samples contained different SNPs and InDels, so the cpSNPs and cpInDels could be used for the identification of oil-tea camellia species and their phylogenetic relationships, and they could also be developed into molecular markers.

The phylogenetic relationships of some plants, including 45 species of the *Taxodiaceae* family [11], 20 species of 20 genera of the *Schisandraceae* family [21], 63 species of the *Populus* genus [22] and 42 species of the *Camellia* genus [23], have been researched, and hypothetical results have been obtained. The phylogenetic trees of the different oil-tea camellia samples were constructed based on the CDSs and non-CDSs of the chloroplast genome in this paper, and the different species of oil-tea camellia and different populations of *C. gaochauensis* were clearly distinguished from each other. Furthermore, the phylogenetic relationships of all samples were illustrated, and the phylogenetic trees were consistent with those previously constructed based on full cpDNAs by our research group [5]. The following main conclusions were reached once again. *C. gauchowensis* and *C. vietnamensis* were merged into the same species, and the undetermined species from Hainan Province was *C. vietnamensis*. Additionally, verifying that *C. osmantha* is an independent species will require additional genetic evidence [24,25]. Moreover, because the samples of species from the same sections of the *Camellia* genus did not belong to the same subclades, divisions of oil-tea camellia species into different sections based on morphological characteristics might be unreasonable [26]. All these results proved that the cpSNPs and cpInDels could be used for the identification of species, populations and phylogenetic relationships in this genus.

Studies analysing the phylogenetic relationships among species or cultivars, constructing genetic maps and identifying the types of germplasm based on SNP or InDel molecular markers from the full chloroplast genomes or nuclear genomes of other plants, such as avocado (*Persea americana*), apple (*Malus pumila*), grape (*Vitis vinifera*), peach (*Prunus persica*) and yellow lupin (*Lupinus luteus*), have had desirable outcomes [27,28,29,30,31]. Therefore, the cpSNPs and cpInDels screened in this paper could be developed into molecular markers for the identification of species and populations, the analysis of genetic diversity, the construction of matrilineages and for research on phylogenetic relationships.

## 5. Conclusions

There were SNPs and InDels in the cpDNAs of all samples. They showed a preference for transition and deletion mutations, respectively. The cpSNPs in the CDSs caused synonymous mutations, missense mutations and gain or loss of the termination codon, and the cpSNPs and cpInDels in the non-CDSs caused variations in gene expression. Both cause the phenotypes of different oil-tea camellia species or different populations of *C. gaochauensis* to differ through mutations in cpDNA genes, so both have certain effects on the evolution of oil-tea camellia species or populations. The cpSNPs and cpInDels were unique to species, and can be used for research on the detection and identification of genetic diversity, species and phylogenetic relationships. These differences can also be developed into molecular markers. The phylogenetic trees based on the CDSs and non-CDSs of cpDNAs indicate again that the undetermined species from Hainan Province is *C. vietnamensis*, and that *C. gauchowenesis* and *C. vietnamensis* must be merged into the same species. Additionally, the genetic relationship among *C. osmantha*, *C. gauchowenesis* and *C. vietnamensis* is much closer than the others, and determining whether *C. osmantha* is an independent species needs more genetic evidence. Meanwhile, the sectional division of the *Camellia* genus based on morphological characteristics may need to be readjusted on the basis of differences in cpDNAs.

## Figures and Tables

**Figure 1 genes-14-01083-f001:**
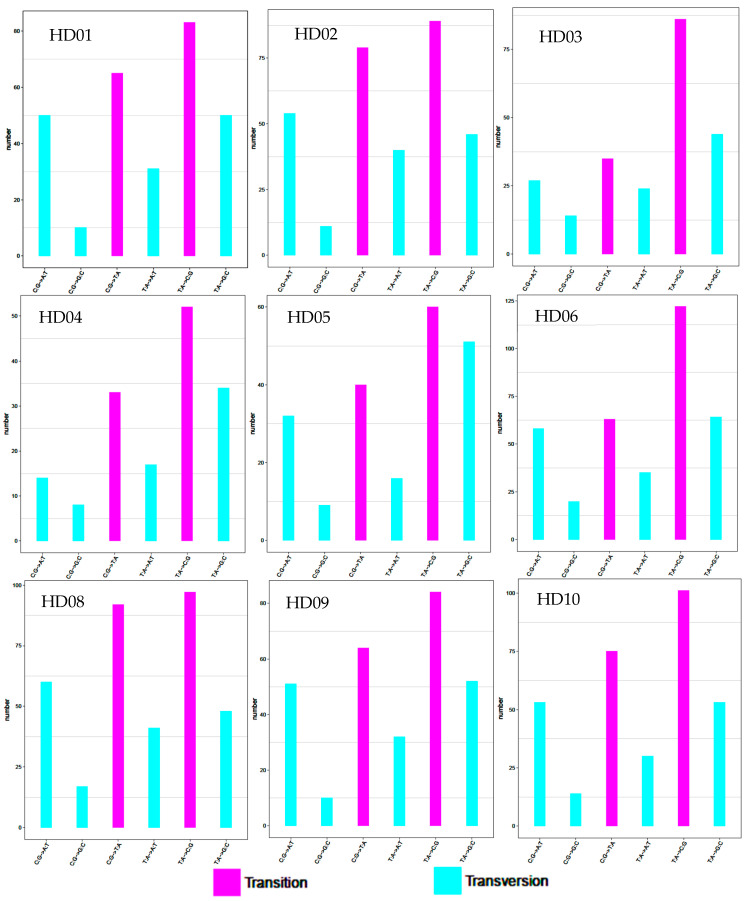
SNP mutations.

**Figure 2 genes-14-01083-f002:**
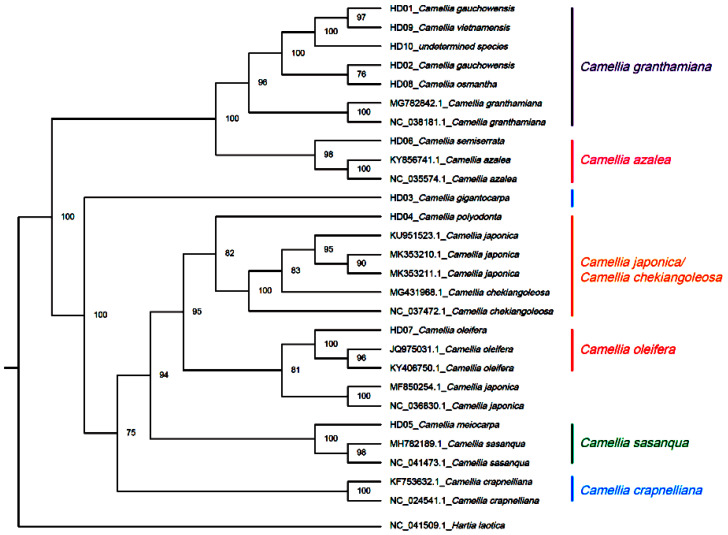
Phylogenetic tree of 13 samples based on CDS using the IQTREE method (ML).

**Figure 3 genes-14-01083-f003:**
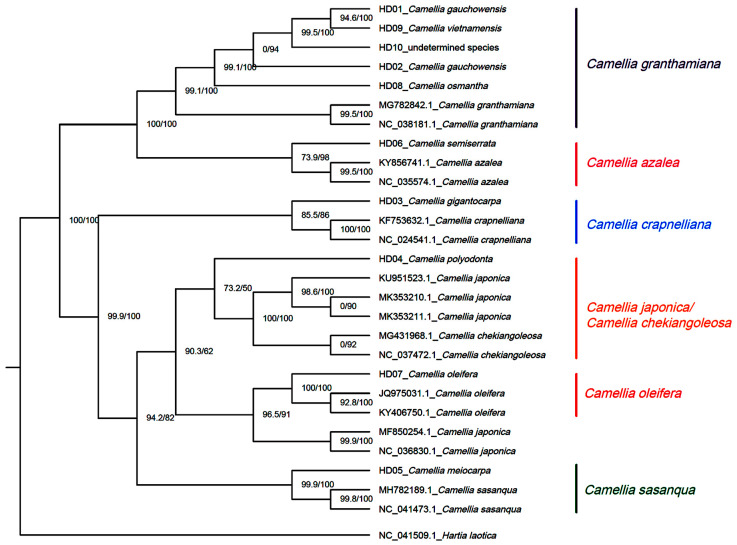
Phylogenetic tree of 13 samples based on non-CDS using the IQTREE method (ML).

**Table 1 genes-14-01083-t001:** The results of the SNP genotyping.

Samples	HD01	HD02	HD03	HD04	HD05	HD06	HD08	HD09	HD10
Homozygous SNP	143	152	80	57	77	129	156	142	128
Heterozygous SNP	3	15	70	44	54	104	43	9	70
Total	146	167	150	101	131	233	199	151	198

**Table 2 genes-14-01083-t002:** The distributions of the SNPs in the genes of the populations of *Camellia gauchowensis.*

Samples	Regions of Gene	Gene (Numbers of the SNPs)	Total
HD01	Downstream	*psbA*(3), *rps16*(1), *psbI*(15), *atpA*(2), *atpI*(2), *petN*(4), *ndhJ*(5)	32
	Exon	*psbA*(1), *Matk*(2), *psbK*(2), *rps2*(1), ***rpoC2***(5), ***rpoB***(5), *psaA*(1), *ndhJ*(1), *atpB*(3), ***rbcL**(1), **accD***(1), ***accD***(3), ***ycf4***(2)	28
	Intergenic space	*rpoB*(1), *psbM*(2), *rpl20*(43)	46
	Intron	*rps16*(2), *atpF*(1), *rpoC1*(2), *ycf3*(1)	6
	Upstream	*rps16*(3), *psbK*(1), *psbI*(2), *rpoB*(3), *psbM*(1), *psbD*(1), *rps4*(1), *ndhC*(2), *atpB*(2), *rpl20*(3)	19
	Upstream; downstream	*pabA;matK*(1)/*psbI*; *psbK*(5)/*atpH*; *atpI*(2)/*psaI*; *accD*(1)*/cemA*; *ycf4*(2)/*petG*; *petL*(1)/*psaJ*; *petG*(1)/*psaJ*(1)/*rpl20*; *rps18*(1)	15
	Total		146
HD02	Downstream	*psbA*(3), *rps16*(1), *psbI*(15), *atpA*(2), *atpI*(2), *petN*(3), *ndhJ*(6), *psaB*(2), *rpl20*(1)	35
	Exon	*matk*(1), *psbK*(2), *rps2*(1), ***rpoC2***(3), *rpoC2*(1), ***rpoB***(4), *rpoB*(2), *psaA*(1), ***ndhJ***(1), *ndhJ*(3), ***atpE***(1), *atpE*(3), *atpB*(3), ***rbcL***(1),***accD***(1), ***accD***(1), *accD*(2), *ycf3*(1), ***psbF***(1), ***psbF***(1), *psbF*(1)	34
	Intergenic space	*psbI*(1), *psbM*(2), *rpl20*(52)	55
	Intron	*rps16*(2), *atpF*(1), *rpoC1*(2), *ycf3*(1)	6
	Upstream	*rps16*(2), *psbK*(1), *psbI*(2), *rpoB*(5), *petN*(1), *psbM*(2), *rps4*(1), *ndhC*(2), *atpB*(2), *petG*(1), *rpl20*(3)	22
	Upstream; downstream	*pabA*; *matK*(1)/*psbI*; *psbK*(4)/*atpA*; *atpF*(1)/*atpH*; *atpI*(2)/*psaI*; *accD*(1)*/cemA*; *ycf4*(2)/*petG*; *petL*(1)/*psaJ*; *petG*(1)/*psaJ*(1)/*rpl20*; *rps18*(1)	15
	Total		167
HD13, HD10 –HD12	Downstream	*psbA*(3), *rps16*(1), *psbI*(12), *atpA*(2), *atpI*(2), *petN*(4), *ndhJ*(5)	29
	Exon	*psbA*(1), *matk*(2), *psbK*(2), *rps2*(1), ***rpoC2***(4), *rpoC2*(1), ***rpoB***(3), *rpoB*(2), ***psbC***(1), *psbC*(4), *psaA*(1), ***ndhJ***(2), *ndhJ*(6), ***atpE***(1), *atpE*(4), *atpB*(3), ***rbcL***(1), ***accD***(1), ***accD***(1), *accD*(1), ***ycf4***(1), *ycf4*(1), ***psbJ***(2), *psbJ*(1), ***psbL***(3), ***psbF***(5), ***psbF***(1), ***psbE***(3), *psbE*(1), ***petL***(4), ***petL***(1), *petG*(1)	66
	Intergenic space	*rpoB*(1), *psbM*(2), *rpl20*(57)	60
	Intron	*rps16*(2), *atpF*(1), *rpoC1*(2), *ycf3*(1)	6
	Upstream	*rps16*(3), *psbK*(1), *psbI*(2), *rpoB*(3), *psbM*(4), *psbD*(1), *rps4*(1), *ndhC*(1), *atpB*(1), *rpl20*(3)	20
	Upstream; downstream	*pabA;matK*(1)/*psbI*; *psbK*(5)/*atpH*; *atpI*(2)/*psaI*; *accD*(1)*/cemA*; *ycf4*(2)/*psaJ*; *petG*(4)/*psaJ*(1)/*rpl20*; *rps18*(1)	17
	Total		198
HD09	Downstream	*psbA*(3), *rps16*(1), *psbI*(15), *atpA*(2), *atpI*(2), *petN*(4), *ndhJ*(5)	32
	Exon	*psbA*(1), *Matk*(2), *psbK*(2), *rps2*(1), ***rpoC2***(4), *rpoC2*(1), ***rpoB***(3), *rpoB*(2), *psaA*(1), *ndhJ*(2), *atpB*(3), ***rbcL***(1), ***accD***(1), ***accD***(1), *accD*(2), ***ycf4***(1), *ycf4*(1)	29
	Intergenic space	*rpoB*(1), *psbM*(2), *rpl20*(49)	52
	Intron	*rps16*(2), *atpF*(1), *rpoC1*(2), *ycf3*(1)	6
	Upstream	*rps16*(3), *psbK*(1), *psbI*(2), *rpoB*(3), *psbM*(1), *psbD*(1), *rps4*(1), *ndhC*(1), *atpB*(1), *rpl20*(3)	17
	Upstream; downstream	*pabA;matK*(1)/*psbI*; *psbK*(5)/*atpH*; *atpI*(2)/*psaI;accD*(1)*/cemA*; *ycf4*(2)/*petG*; *petL*(1)/*psaJ; petG*(1)/*psaJ*(1)/*rpl20*; *rps18*(1)	15
	Total		151
HD08	Downstream	*psbA*(3), *rps16*(2), *psbI*(20), *atpA*(3), *atpI*(2), *petN*(7), *ndhJ*(5), *petA*(1)	43
	Exon	*matk*(1), *psbK*(2), *rps2*(1), ***rpoC2***(3), *rpoC2*(2), ***rpoB***(3), *rpoB*(2), ***psaA***(1), *psaA*(2), ***ndhJ***(2), ***atpE***(4), *atpE*(5), *atpB*(3), ***rbcL***(1), ***accD***(1), ***accD***(1), *accD*(2), *ycf4*(1), ***psbF***(1), *psbF*(1), *petG*(1), *rpl20*(1)	40
	Intergenic space	*psbM*(4), *rps4*(1), *rpl20*(60)	65
	Intron	*rps16*(3), *atpF*(1), *rpoC1*(3), *ycf3*(3)	10
	Upstream	*rps16*(3), *psbK*(1), *psbI*(2), *rpoB*(3), *psbM*(2), *rps4*(1), *ndhC*(1), *atpB*(2), *rpl20*(5)	20
	Upstream; downstream	*pabA*; *matK*(1)/*psbI*; *psbK*(4)/*atpH*; *atpI*(2)/*rpoC2*; *rpoC1*(1) /*ndhJ*; *ndhC*(3)*/psaI; accD*(1)*/cemA*; *ycf4*(2)/*petA*; *cemA*(2)/*petG*; *petL*(1)/*psaJ; petG*(1)/*psaJ*(1)/*rpl33*(1)/*rpl20*; *rps18*(1)	21
	Total		199
HD03	Downstream	*psbA*(1), *psbI*(12), *atpA*(1), *atpI*(1), *petN*(7), *petA*(3)	25
	Exon	***matk***(1), *psbK*(1), ***atpF***(1), *rps2*(1), *rpoC2*(1), ***rpoC1***(1), ***rpoB***(2), ***psbC***(1), *psbC*(3), ***ndhJ***(1), ***ndhJ***(2), *ndhJ*(8), ***atpE***(3), *atpE*(5), ***atpB***(2), *atpB*(1), ***rbcL***(1), ***accD***(2), *ycf4*(1), ***psbJ***(3), *psbJ*(2), ***psbL***(3), ***psbF***(1), ***psbF***(1), ***petL***(4), *petL*(2), *petG*(2)	56
	Intergenic space	*rps4*(2), *psbM*(2), *rpl20*(31)	35
	Intron	*rps16*(2), *atpF*(2), *ycf3*(2)	6
	Upstream	*rps16*(1), *psbK*(1), *psbI*(2), *rpoB*(1), *psbM*(1), *ndhC*(1), *rpl20*(1)	8
	Upstream; downstream	*psbI; psbK*(1)/*ndhJ*; *ndhC*(2)/*psaI*; *accD*(1)/*psbJ*; *psbE*(2)/*psaJ*; *petG*(13)/*psaJ*(1)	20
	Total		150
HD04	Downstream	*psbA*(1), psbI(12), *atpI*(1), *petN*(2), *atpB*(1)	17
	Exon	*psbK*(1), *rpoC*1(1), ***rpoB***(3), *rpoB*(1), ***ndhJ***(4), ***ndhJ***(1), *ndhJ*(7), ***atpE***(2), *atpE*(6), ***atpB***(2), *ycf4*(1), ***psbJ***(3), *psbJ*(2), ***psbL***(3), ***psbF***(1), ***psbF***(1), *psbF*(1)	40
	Intergenic space	*psbM*(1), *rpl20*(19)	20
	Intron	*rps16*(2), *atpF*(1), *ycf3*(1)	4
	Upstream	*rps16*(1), *psbI*(1), *atpF*(1), *rpoB*(1), *psbM*(5), *rps4*(1), *petG*(1), *rpl20*(1)	12
	Upstream; downstream	*psbI*; *psbK*(1)/*ndhJ*; *ndhC*(2)/*ndhC*; *atpE*(1)/*psaJ*; *accD*(1)/*psaJ*; *petG*(2)/*psaJ*(1)	8
	Total		101
HD05	Downstream	*psbA*(1), psbI(12), *atpI*(1), *petN*(2), *atpB*(1)	17
	Exon	***matK***(2), ***atpF***(1), *rps2*(1), *rpoC*2(2), ***rpoB***(2), *rpoB*(1), *psbC*(2), *psaB*(1), ***ndhJ***(4), *ndhJ*(6), ***atpE***(1), *atpE*(4), ***atpB***(1), *ycf4*(1), ***psbJ***(2), *psbJ*(1), ***psbL***(3), ***psbF***(1), ***psbF***(1), *psbF*(1), ***petL***(4), *petL*(2), *petG*(2)	47
	Intergenic space	*psbI*(1), *psbM*(1), *rpl20*(39)	41
	Intron	*rps16*(2), *atpF*(1), *ycf3*(1)	4
	Upstream	*rps16*(1), *psbK*(1), *psbI*(2), *rpoB*(1), *ndhC*(3), *atpB*(1), *rpl20*(1)	10
	Upstream; downstream	*psbA*; *matk*(1)/*psbI*; *psbK*(2)/*atpF*(2)/*psaI*; *accD*(1)/*psaJ*; *petG*(5)/*psaJ*(1)	12
	Total		131
HD06	Downstream	*psbA*(3), psbI(15), *atpA*(2), *atpI*(1), *petN*(7), *ndhJ*(4)	32
	Exon	*matK*(1), *psbK*(2), *atpH*(1), *rps20*(1), ***rpoC*****2**(1), *rpoC*2(1), ***rpoB***(3), *rpoB*(3), *psbC*(1), *psbC*(4), ***ndhJ***(1), ***ndhJ***(4), *ndhJ*(7), ***atpE***(6), *atpE*(9), ***atpB***(5), *atpB*(6), ***rbcL***(1), ***accD***(1), *accD*(1), *ycf4*(1), ***psbJ***(2), *psbJ*(1), ***psbL***(1), ***psbF***(1), ***psbF***(1), *psbF*(1), ***psbE***(2), ***psbE***(1), ***petL***(4), ***petL***(1), *petL*(2), *petG*(2), *rps18*(1)	80
	Intergenic space	*psbM*(3), *rpl20*(50)	53
	Intron	*rps16*(2), *atpF*(1), *rpoC1*(1), *ycf3*(1)	5
	Upstream	*matK*(1), *rps16*(3), *psbK*(1), *psbI*(2), *rpoB*(3), *psbM*(6), *rps4*(1), *ndhC*(1), *atpB*(2), *petG*(1), *rpl20*(2)	23
	Upstream; downstream	*psbI*; *psbK*(4)/*atpH*; *atpI*(2)*/psaA*; *ycf3*(1)*/ndhJ*; *ndhC*(2)/*psaJ*; *accD*(1)/*cemA*; *ycf4*(2)/*psaI*(1)/*petG*; *petL*(4)*/petG*(1)/*psaJ*; *petG*(20)*/psaJ*(1)*/rpl20*; *rps18*(1)	40
	Total		233

Note: The terms “upstream” and “downstream” are defined as 1 kb away from the transcription start site and transcription end site, respectively, taking into account the strand of the mRNA. If a variant is located in both downstream and upstream regions (possibly for 2 different genes), then “upstream; downstream” will be printed as the output. The intergenic space is >1 kb away from any gene of both. Gene names shown in the bold font and green, blue and red indicate a nonsynonymous mutation, stop-gain mutation and stoploss mutation, respectively. The chloroplast genome of *C. oleifera* was used as the reference genome. The same as below.

**Table 3 genes-14-01083-t003:** The results of the annotations of SNPs in exons.

Samples	HD01	HD02	HD03	HD04	HD05	HD06	HD08	HD09	HD10
Total	28	34	56	40	47	80	40	29	66
Nonsynonymous SNV	11	12	27	18	21	32	16	11	31
Stopgain SNV	1	2	2	2	1	2	1	1	2
Stopiloss SNV	-	-	-	-	-	1	-	-	1
Synonymous SNV	16	20	27	20	25	45	23	17	32

**Table 4 genes-14-01083-t004:** The distributions of the same SNPs in the genes of the different sample groups.

	Regions of Gene	Gene (Numbers of the SNPs)	Total
HD01 HD02 HD13	Downstream	*psbA*(3), *rps16*(1), *atpA*(2), *atpI*(2)	18
	Exon	*matk*(2), *rps2*(1), *psaA*(1), *atpB*(3), ***rbcL***(1), ***accD***(1)	9
	Intergenic space	*rpoB*(1), *psbM*(2)	3
	Intron	*rps16*(2), *atpF*(1), *rpoC1*(2), *ycf3*(1)	6
	Upstream	*psbK*(1), *psbI*(2), *rps4*(1), *rpl20*(3)	7
	Upstream; downstream	*pabA*; *matK*(1)/*atpH*; *atpI*(2)/*psaI*; *accD*(1)*/cemA*; *ycf4*(2)/*psaJ*(1)/*rpl20*; *rps18*(1)	8
	Total		51
HD01 HD09 HD10– HD13	Downstream	*psbA*(3), *rps16*(1), *atpA*(2), *atpI*(2), *petN*(4), *ndhJ*(5)	17
	Exon	*psbA*(1), *matk*(2), *psbK*(2), *rps2*(1), *rpoC2*(5), *rpoB*(5), *atpB*(3), ***rbcL***(1), ***accD***(1), *ycf4*(2)	23
	Intergenic space	*rpoB*(1), *psbM*(2)	3
	Intron	*rps16*(2), *atpF*(1), *rpoC1*(2), *ycf3*(1)	6
	Upstream	*rps16*(3), *psbK*(1), *psbI*(2), *rpoB*(3), *psbD*(1), *rps4*(1), *rpl20*(3)	14
	Upstream; downstream	*pabA*; *matK*(1)/*psbI*; *psbK*(5)/*atpH*; *atpI*(2)/*psaI*; *accD*(1)*/cemA*; *ycf4*(2)/*psaJ*(1)/*rpl20*; *rps18*(1)	13
	Total		76
HD01 HD08 HD09	Downstream	*psbA*(3), *atpI*(2), *ndhJ*(5)	10
	Exon	*rps2*(1), *rpoC2*(5), *rpoB*(5), *atpB*(3), ***rbcL***(1), ***accD***(1), *ycf4*(2)	18
	Intron	*atpF*(1), *ycf3*(1)	2
	Upstream	*rps16*(3), *psbK*(1), *psbI*(2), *rpoB*(3), *rps4*(1), *atpB*(1)	11
	Upstream; downstream	*psaI*; *accD*(1)*/cemA*; *ycf4*(2)/*petG*; *petL*(1)/*psaJ*; *petG*(1)/*psaJ*(1)/*rpl20*; *rps18*(1)	7
	Total		48
HD01 HD02 HD08 HD09 HD10– HD13	Downstream	*psbA*(3), *atpI*(2)	5
	Exon	*rps2*(1), *atpB*(3), ***rbcL***(1), ***accD***(1)	6
	Intron	*atpF*(1), *ycf3*(1)	2
	Upstream	*psbK*(1), *psbI*(2), *rps4*(1)	4
	Upstream; downstream	*cemA*; *ycf4*(2)/*psaJ*(1)/*rpl20*; *rps18*(1)	4
	Total		21
HD03 HD05	Downstream	*psbA*(1), *psbI*(12), *atpA*(1)	14
	Exon	***matk***(1), ***atpF***(1), *rps2*(1), *ycf4*(1), *petL*(6), *petG*(2), ***psbF***(1), ***psbF***(1)	14
	Intron	*rps16*(2)	2
	Upstream	*rps16*(1), *psbK*(1), *psbI*(2), *rpoB*(1), *rpl20*(1)	6
	Upstream; downstream	*psaI*; *accD*(1)/*psaJ*(1)	2
	Total		38
HD04 HD06	Downstream	*atpI*(1)	1
	Exon	***rpoB***(3), ***ndhJ***(1), ***ndhJ***(4), *ndhJ*(7), ***psbF***(1), ***psbF***(1), *psbF*(1)	18
	Intron	*rps16*(2), *atpF*(1), *ycf3*(1)	4
	Upstream	*rps4*(1), *petG*(1)	2
	Upstream; downstream	*psaJ*; *accD*(1)/*psaJ*(1)	2
	Total		27
	Intron	*rps16*(2)	2
	Upstream; downstream	*accD*(1)/*psaJ*(1)	1
	Total		3
All samples	Total		0

Note: Gene names shown in the bold font and green, blue and red indicate a nonsynonymous mutation, stop-gain mutation and stoploss mutation, respectively.

**Table 5 genes-14-01083-t005:** The results of the identification of InDels.

Samples	HD01	HD02	HD03	HD04	HD05	HD06	HD08	HD09	HD10
Deletion InDels	22	28	25	13	17	22	25	21	20
Insertion InDels	17	17	13	8	7	16	18	17	16
Total	39	45	38	21	24	38	43	38	36
The ratio of deletion to insertion	1.29	1.65	1.92	1.63	2.43	1.38	1.39	1.24	1.25

**Table 6 genes-14-01083-t006:** The distributions of the InDels in the genes.

Samples	Regions of Gene	Gene (Numbers of the SNPs)	Total
HD01	Downstream	*psbA*(1), *rps16*(1), *psbI*(3), *atpA*(2), *psbZ*(1), *psaB*(1), *petA*(1)	10
	Intergenic space	*psbM*(1), *rps4*(1), *rpl20*(4)	6
	Intron	*rps16*(2), *ycf3*(1)	3
	Upstream	*matK*(2), *psbK*(1), *psbD*(1), *ycf3*(1), *rps4*(1), *ndhC*(1)	7
	Upstream; downstream	*psbI*; *psbK*(1)/*atpI*(1)/*psaI*; *accD*(1)/*psaI*(1)*/ycf4*; *accD*(1)*/cemA*; *ycf4*(3)/*petA*; *cemA*(1)/*psaJ*(1)/*rpl33*(1)/*rpl20*; *rps18*(2)	13
	Total		39
HD02	Downstream	*psbA*(1), *rps16*(1), *psbI*(3), *atpA*(4), *psbZ*(1), *psaB*(3), *petA*(2)	15
	Intergenic space	*psbM*(1), *rps4*(1), *rpl20*(6)	8
	Intron	*rps16*(2), *ycf3*(1)	3
	Upstream	*matK*(2), *rps16*(1), *psbK*(1), *psbD*(1), *ycf3*(1), *rps4*(1), *ndhC*(1)	8
	Upstream; downstream	*psbA*; *matK*(1)/*psbI*; *psbK*(1)/*atpI*(1)/*ndhJ*; *ndhC*(1)/*psaI*; *accD*(1)/*ycf4*; *accD*(1)*/cemA*; *ycf4*(1)/*petA*; *cemA*(1)/*psaJ*(1)/*rpl33*(1)/*rpl20*; *rps18*(1)	11
	Total		45
HD03	Downstream	*psbA*(1), *psbI*(2), *atpA*(3), *atpI*(1), *psaB*(1), *ndhJ*(1), *petA*(3)	12
	Exon	*rpoC2*(1)	1
	Intergenic space	*psbM* (2), *rps4*(1), *rpl20*(4)	7
	Upstream	*matK*(1), *rps16*(2), *psbK*(1), *yfc3*(1), *rps4*(2), *ndhC*(2), *petG*(1)	10
	Upstream; downstream	*psbI*; *psbK*(1)/*psaI*; *accD*(1)/*cemA*; *ycf4*(2)/*petA*; *cemA*(1)/*psaJ*(1)/*rpl33*(1)/ *rpl20*; *rps18*(1)	8
	Total		38
HD04	Downstream	*psbI*(2), *atpA*(3), *psaB*(1)	6
	Intergenic space	*rpl20*(3)	3
	Upstream	*ndhC*(2)	2
	Upstream; downstream	*atpI*(1)/*psaA*; *ycf3*(1)/*ycf3*; *rps4*(1)*/psaI*; *accD*(1)/*cemA*; *ycf4*(2)/*petA*; *cemA*(1)/*psaJ*(1)/*rpl33*(1)/*rpl20*; *rps18*(1)	10
	Total		21
HD05	Downstream	*psbA*(1), *psbI*(2), *atpA*(3), *psaB*(1)	7
	Intergenic space	*rpl20*(5)	5
	Upstream	*rps16*(2), *ndhC*(1)	3
	Upstream; downstream	*psbI*; *psbK*(1)/*atpI*; *rps2*(1)/*psaI*; *accD*(1)/*cemA*; *ycf4*(2)/*petA*; *cemA*(1)/*psaJ*(1)/*rpl33*(1)/*rpl20*; *rps18*(1)	9
	Total		24
HD06	Downstream	*psbA*(1), *psbI*(3), *atpA*(2), *psbZ*(1), *psaB*(2), *petA*(1)	10
	Intergenic space	*psbM*(1), *rps4*(2), *rpl20*(7)	10
	Intron	*rps16*(2), *ycf3*(1)	3
	Upstream	*matK*(4), *ycf3*(1), *ndhC*(2)	7
	Upstream; downstream	*psaI*; *accD*(1)/*ycf4*; *accD*(1)*/cemA*; *ycf4*(2)/*petA*; *cemA*(1)/*psaJ*(1)/*rpl33*(1)/ *rpl20*; *rps18*(1)	8
	Total		38
HD08	Downstream	*psbA*(1), *rps16*(2), *psbI*(3), *atpA*(2), *psbZ*(1), *psaB*(2), *petA*(2)	13
	Intergenic space	*psbM*(1), *rps4*(1), *rpl20*(7)	9
	Intron	*rps16*(2), *ycf3*(1)	3
	Upstream	*matK*(2), *rps16*(1), *psbK*(1), *petN*(1), *psbM*(1), *psbD*(1), *ycf3*(1), *ndhC*(1)	9
	Upstream; downstream	*psbI*; *psbK*(1)/atpH; atpI(1)/atpI(1)/*psaI*; *accD*(1)/*ycf4*; *accD*(1)*/cemA*; *ycf4*(2)/*petA*; *cemA*(1)/*rpl33*(1)/*rpl20*; *rps18*(1)	9
	Total		43
HD09	Downstream	*psbA*(1), rps16(1), *atpI*(3), *atpA*(1), *psbZ*(1), *psaB*(1), *petA*(1)	9
	Intergenic space	*psbM* (1), *rps4*(1), *rpl20*(4)	6
	Intron	*rps16*(2), *ycf3*(1)	3
	Upstream	*matK*(2), *psbK*(1), *psbD*(1), *ycf3(1)*, *rps4*(1), *ndhC*(1)	7
	Upstream; downstream	*psbI*; *psbK*(1)/*atpI*(1)/*psaI*; *accD*(1)/*psaI*(1)/*ycf4*; *accD*(1)/*cemA*; *ycf4*(3)/*petA*; *cemA*(1)/*psaJ*(1)/*rpl33*(1)/*rpl20*; *rps18*(2)	13
	Total		38
HD10	Downstream	*psbI*(1), *rps16*(1), *psbI*(1), *atpA*(2), *psbZ*(1), *psaB*(1), *petA*(1)	8
	Intergenic space	*psbZ*(1), *rps4*(1), *rpl20*(4)	6
	Intron	*rps16*(1), *ycf3*(1)	2
	Upstream	*matK*(2), *psbK*(1), *psbD*(1), *ycf3*(1), *rps4*(1), *ndhC*(1)	7
	Upstream; downstream	*psbI*; *psbK*(1)/*atpI*(1)/*psaI*; *accD*(1)/*psaI*(1)*/ycf4*; *accD*(1)/*cemA*; *ycf4*(3)/*petA*; *cemA*(1)/*psaJ*(1)/*rpl33*(1)/*rpl20*; *rps18*(2)	13
	Total		36

Note: The bolding of a gene’s name indicates a non-frameshift deletion.

**Table 7 genes-14-01083-t007:** The comparisons of the InDels in the intergenic space of *rpl20* genes of the different samples.

Samples	InDels (Insertion/Deletion)	Total	Samples	InDels (Insertion/Deletion)	Total
HD01 HD09– HD13	TATAATA (Deletion)	4	HD05	A (Insertion)	5
	AATAT (Deletion)			TATAATA (Deletion)	
	T (Deletion)			CTTAC (Deletion)	
	T (Deletion)			T (Deletion)	
				TCTTTT (Deletion)	
HD02	TATAATA (Deletion)	6	HD06	TATAATA (Deletion)	7
	TCATGAT (Deletion)			AATAT (Deletion)	
	AT (Insertion)			AT (Insertion)	
	AATAT (Deletion)			T (Deletion)	
	T (Deletion)			T (Insertion)	
	T (Deletion)			TATGGTAATCCA (Deletion)	
				T (Deletion)	
HD03	TATAATA (Deletion)	4	HD08	TATAATA (Deletion)	7
	A (Insertion)			T (Insertion)	
	A (Deletion)			AATAT (Deletion)	
	T (Deletion)			T (Deletion)	
HD04	TATAATA (Deletion)	3		T (Insertion)	
	AATAG (Insertion)			TATGGTAATCCA (Deletion)	
	T (Deletion)			T (Deletion)	

## Data Availability

The data presented in this study are available on request from the corresponding author. The data are not publicly available due to privacy.

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
