# Peer review of "Comparisons of Chloroplast Genome Mutations among 13 Samples of Oil-Tea Camellia from South China"

_genes, 2023, doi:10.3390/genes14051083_

Round 1

Reviewer 1 Report

The article contains interesting data that reveal a new approach to the genetic differentiation of different species of oil-tea camellia. Although individual plants mainly from Hainan province were examined, the meaning of the study is broader.

I have the following question: is there a specific combination of mutations that can determine a given plant to a species? Please rank the gene mutations (indels) in order of importance, which ones determine how far phylogenetically one species is from another?

Figure 1 is of very poor quality. The text is not readable, it is too small, and it must be edited. Maybe that's where my questions about markers come from, as I can't read the text in Figure 1.

Notes: related to the references. In the text they should be in parentheses, and in the list - in the order of mention in the manuscript. I ask authors to first look at the journal's reference requirements. At Ref. 4 the year is missing. Additionally, authors should check for identical passages with this article, which are quite similar. For instance, the abstract contains the same conclusion as in the paper https://www.ncbi.nlm.nih.gov/pmc/articles/PMC8860168/

As a whole, the English language is fine, there is a recurring misspelling of the in-text citation: e.g. "Chen et al," should be "Chen et al.," and should be followed by the reference in square brackets.

Reviewer 2 Report

Chloroplast sequences were analyzed and compared in various types of plants of the genus Camellia, and it contains good research results that revealed the related relationship between these species. Various SNPs and InDel locus are suggested to be used as data for making various molecular markers in the future.

Authors need a figure legend of Figure1 and need to increase the resolution.

In Figure 2, the yellow color is hard to see, so it seems to be better to change it to another color.

Also, in Tables 2, 3, 4, 5, and 7, it should be clearly expressed that the insertions and deletions were determined based on which reference.

In Tables 2, 4, 6, and 7, the authors need to present the data more concisely.

Round 2

Reviewer 1 Report

 The manuscript is substantially improved.

The English style and grammar use are good.